# Cardiac Toxicity Associated with Cancer Immunotherapy and Biological Drugs

**DOI:** 10.3390/cancers13194797

**Published:** 2021-09-25

**Authors:** Andrea Montisci, Maria Teresa Vietri, Vittorio Palmieri, Silvia Sala, Francesco Donatelli, Claudio Napoli

**Affiliations:** 1Division of Cardiothoracic Intensive Care, ASST Spedali Civili, 25123 Brescia, Italy; montisci.andrea@yahoo.it; 2Department of Precision Medicine, University of Campania “Luigi Vanvitelli”, 81100 Naples, Italy; mtvmail2021@gmail.com; 3Department of Cardiac Surgery and Transplantation, Ospedali dei Colli Monaldi-Cotugno-CTO, 80131 Naples, Italy; vpalmieri68@gmail.com; 4Department of Anesthesia and Intensive Care, University of Brescia, 25121 Brescia, Italy; s.sala011@unibs.it; 5Cardiac Surgery, University of Milan, 20122 Milan, Italy; 6Department of Cardiac Surgery, Istituto Clinico Sant’Ambrogio, 20149 Milan, Italy; 7Clinical Department of Internal Medicine and Specialistics, University Department of Advanced Clinical and Surgical Sciences, University of Campania “Luigi Vanvitelli”, 81100 Naples, Italy; cnapoli2@tin.it; 8IRCCS SDN, 80143 Naples, Italy

**Keywords:** cancer, immunotherapy, myocarditis, trastuzumab, chimeric antigen receptor-modified T (CAR-T), immune checkpoint inhibitors

## Abstract

**Simple Summary:**

Immunotherapy is increasingly being used to treat solid tumors and lymphoproliferative diseases. The main classes of drugs are: HER-2-targeted therapies, CTLA-blockers, PD/PDL-1 inhibitors, CAR-T therapy. All these drugs are associated with meaningful cardiac toxicity, ranging from a transient decline of left ventricular function with complete reversibility to myocarditis with a high fatality rate.

**Abstract:**

Cancer immunotherapy significantly contributed to an improvement in the prognosis of cancer patients. Immunotherapy, including human epidermal growth factor receptor 2 (HER2)-targeted therapies, immune checkpoint inhibitors (ICI), and chimeric antigen receptor-modified T (CAR-T), share the characteristic to exploit the capabilities of the immune system to kill cancerous cells. Trastuzumab is a monoclonal antibody against HER2 that prevents HER2-mediated signaling; it is administered mainly in HER2-positive cancers, such as breast, colorectal, biliary tract, and non-small-cell lung cancers. Immune checkpoint inhibitors (ICI) inhibit the binding of CTLA-4 or PD-1 to PDL-1, allowing T cells to kill cancerous cells. ICI can be used in melanomas, non-small-cell lung cancer, urothelial, and head and neck cancer. There are two main types of T-cell transfer therapy: tumor-infiltrating lymphocytes (or TIL) therapy and chimeric antigen receptor-modified T (CAR-T) cell therapy, mainly applied for B-cell lymphoma and leukemia and mantle-cell lymphoma. HER2-targeted therapies, mainly trastuzumab, are associated with left ventricular dysfunction, usually reversible and rarely life-threatening. PD/PDL-1 inhibitors can cause myocarditis, rare but potentially fulminant and associated with a high fatality rate. CAR-T therapy is associated with several cardiac toxic effects, mainly in the context of a systemic adverse effect, the cytokines release syndrome.

## 1. Introduction

Patients treated with either chemotherapy or radiation have a higher cancer relapse chance, and tumors may gain resistance to treatment. These drawbacks have encouraged the discovery of small molecules, peptides, and monoclonal antibodies for immunotherapeutic applications that stimulate the native immune defense system for cancer treatment [1] (Table 1 and Figure 1).

A class of treatment modalities is the immunotherapies that exploit the immune system’s capabilities in anticancer response [9].

Treatment based on T-cell checkpoint blockade therapy resulted in clinical responses for different types of solid tumors [10].

Initially, immunotherapy studies employed cytokines in order to induce a non-specific upregulation of the immune response. These therapies were, however, been associated with high toxicity and a relatively low response rate, prompting the development of improved immunotherapeutic strategies, such as monoclonal antibody-based treatments.

In 1997, the first Food and Drug Administration (FDA)-approved drug of this class was rituximab, a monoclonal antibody directed against the cluster differentiation (CD)-20, for the treatment of B-cell lymphomas [11].

In the following years, the chimeric antigen receptor T cell therapy (CAR-T) was then developed, a therapy that joins the antigen-binding properties of antibodies with the cytolytic ability of T cells [12]. CAR-T are genetically modified cells with the expression of an extracellular antigen-recognition domain. This allows the modified autologous cells to be redirected to surface antigens on cancer cells for destruction [13].

T cell activation in the immune system is regulated by a balance of co-stimulation and inhibition pathways. Co-inhibition pathway receptors on T cells, including cytotoxic T lymphocyte antigen 4 (CTLA-4) or programmed cell death 1 (PD-1) receptors, can bind to ligands on antigen-presenting cells. The binding to the receptors causes a reduction in the immune response and T cell proliferation. To elude the local immune response, tumor cells overexpress these ligands, leading to proliferation without control [14,15,16,17]. The immune checkpoint inhibitors (ICIs) are receptor antagonist monoclonal antibodies that reactivate the anticancer response of native T cells [9]. Several antibodies targeting cellular immune checkpoints (PD-1/PD-L1 and CTLA-4) have been developed to determine the activation of T cells and subsequent tumor control. This treatment strategy is effective in tumors with a high mutation load [18,19,20,21,22,23].

However, many patients report resistance over time that leads to the use of combination CTLA-4 and PD-1 antagonist treatments. These novel therapies have increased significant systemic adverse effects, which can affect multiple organs, such as the gastrointestinal tract, lungs, endocrine, musculoskeletal, renal, nervous, hematologic systems, and skin. An increase in cardiovascular toxicities, which, although rare, are potentially fatal complications has been described [24].

## 2. Drugs

### 2.1. Trastuzumab and HER-2 Targeted Therapies

Trastuzumab deruxtecan (ENHERTU), a biologic antineoplastic agent approved by the FDA in 1998, was among the first available targeted chemotherapies (Table 1). It is a monoclonal antibody against human epidermal growth factor receptor 2 (HER2); it prevents HER2-mediated signaling by binding to an extracellular domain of this receptor that inhibits HER2 homodimerization [25]. Moreover, it is thought to facilitate antibody-dependent cellular cytotoxicity, leading to cell death that expresses HER2 [26].

The use of trastuzumab resulted in more prolonged survival in breast cancer patients with HER2-positive breast cancer. It is used in the therapy of HER2-positive breast cancer as an associated therapy in combination with anthracycline or taxane-based chemotherapy and in metastatic, HER2-positive, breast cancer as a monotherapy or associated with paclitaxel. In the US, it is also used for gastric tumors in association with cisplatin [2].

In recent years in some patients, trastuzumab resistance has been observed, thus for treatment of HER2 positive breast cancer patients with trastuzumab resistance, other drugs were produced [27].

In patients with HER2-positive gastric cancer, the combination of trastuzumab with first-line chemotherapy, has shown an improvement in survival, becoming the standard-of-care treatment. In patients with other solid tumors presenting HER2 overexpression, such as colorectal, biliary tract, non-small-cell lung, and bladder cancers, other HER2-targeted therapies are also being evaluated [28].

#### Cardiotoxicity of Trastuzumab

Trastuzumab is known to cause cardiotoxicity, mostly represented by a reduction in left ventricular function (LVEF) [29].

Studies reporting cardiac toxicity from HER2-targeted therapies are summarized in Table 2. As illustrated in Figure 2, trastuzumab may cause endothelial dysfunction and, by doing so, it may induce microvascular disease, myocardial edema, and parcellar necrosis leading to myocardial dysfunction, reduced ventricular systolic shortening, and increased ventricular filling pressure, which is the hallmark of the heart failure phenotype. However, additional mechanisms of cardiac toxicity associated with trastuzumab are hypothesized and largely unknown, as discussed later.

The risk of cardiotoxicity is major in patients subjected to associated anthracycline therapy. The cardiotoxicity is usually reversible with the discontinuation of treatment [44]. However, it can progress to become a clinically significant cardiac failure with myocardial dysfunction [45].

By the time trastuzumab was approved in 1998 for the treatment of HER2-positive metastatic breast cancer, cardiovascular toxicity was identified as a potentially frequent untoward event, with an incidence between 8% and 30% depending on treatment strategies with trastuzumab alone or trastuzumab in association with anthracycline [31]. More recent data have been consistent with earlier reports on the association of trastuzumab as adjuvant therapy for early-stage and locally advanced HER2-positive breast cancer with pauci-symptomatic decline in the left ventricular (LV) ejection fraction (EF), a measure of LV chamber systolic function, and with overt heart failure [46]. Because the definition of cardiac toxicity based on the threshold of LVEF reduction has been heterogeneous in several trials on trastuzumab and the decline in LFEF was seen with a number of different combinations of trastuzumab with additional anticancer agents, an estimate of the direct and specific contribution of trastuzumab to heart dysfunction and/or heart failure remains variable.

With the definition of cardiac dysfunction as LVEF decline of at least 10% below 55% or 50%, or the presentation with overt congestive heart failure, the prevalence of cardiac toxicity was reported to be between 6% and 35%, with overt heart failure reported in 1% to 14% of the cases [47]. Overall, in approximately 10,000 patients, the relative risk of congestive heart failure associated with trastuzumab treatment was estimated at 5.1 (95% confidence interval (CI) 3.0–8.7, *p* < 0.0001). In the real world, where the rate of exclusion of subjects with LVEF below 55%, older than 65 years, or with mediastinal radiation therapy or hormone therapy, the rate of cardiac toxicity was found to range between 11% and 43%, with overt heart failure reported in a range of 0% to 9% [47].

An echocardiographic follow-up in patients treated with trastuzumab may be useful to detect LVEF decline and then withdraw trastuzumab, or avoid treatment in those with baseline LVEF < 50% at baseline; evaluation of LVEF should be performed at 4 or 6 weeks [48] and repeated according to findings, taking into account that LVEF is a measure of LV chamber systolic function with a significant intra and inter-observer variability [49], and affected by loading conditions. Changes in LVEF do not ever specifically represent changes in systolic myocardial function [50].

Echocardiographic surveillance of patients treated with trastuzumab demonstrated that the reduction in LVEF or heart failure may be reversed with trastuzumab withdrawal and the introduction of selective beta-1 receptor blockers and angiotensin-I converting enzyme inhibitors [51]. Moreover, the incidence of cardiac toxicity with trastuzumab was not found to be related to cumulative dose or treatment duration with trastuzumab [52]. In addition, after trastuzumab suspension and LVEF recovery and/or heart failure resolved, the reintroduction in trastuzumab is not invariably related to a new decline in LFEF or heart failure presentation [51,53,54]. Following troponins during trastuzumab treatment and after trastuzumab withdrawal due to LVEF reduction demonstrated that troponins may not return to baseline, indicating an accelerated myocardial cell turn-over even after recovery and increased susceptibility to cardiac damage with trastuzumab re-challenge [55].

Macroscopic evidence of cardiac dysfunction associated with trastuzumab treatments has not been linked to a specific mechanism. In the adult heart, ErbB tyrosine kinases receptors and contribute to preserving the structure and the function of the myocytes. ErbB receptors activation in the heart is led by ErbB4 ligand neuregulin (NRG)-1, which is, in turn, linked to EGF-like growth factor released by the endothelial cells in normal hearts with normal microcirculation, with cardioprotective effects, at least in vitro and in vivo heart failure models. In cancer, uncontrolled tumor growth is led by gene amplification and overexpression of ErbB2 and ligand-independent ErbB2-ErbB3 heterodimer complex.

Trastuzumab affects NRG-1-induced ErbB signaling, and interferes with cell protective pathways, also through ErbB2 antagonists, as described for lapatinib and pertuzumab [56]. Cardiac myocytes rely on HER2 to achieve enough protection against reactive oxygen species (ROS) [57] and for neo angiogenesis [58]. The association of anthracyclines with increased cardiac toxicity by trastuzumab has have been linked with an anthracycline-induced increment of ROS and oxidative stress [57,59].

Severe cardiac toxicity due to HER2-directed therapies requiring Intensive Care Unit (ICU) admission and advanced therapies is rare and usually reversible [60].

Few cases needing intensive care treatment have been described. Minichillo et al. [61] described a case of cardiac toxicity in a 49-year-old patient who received trastuzumab-based therapy due to metastatic breast cancer. The patient developed severe cardiogenic shock treated with inotropes and intraortic balloon pump, with subsequent slow recovery, finally allowing trastuzumab resumption.

Castells [62] reported a case of LVAD implantation due to cardiogenic shock after adjuvant therapy with trastuzumab/doxorubicin. The patient showed recovery after 4 months, and he had the LVAD explanted. Few additional cases have been described, characterized by myocardial recovery and trastuzumab resumption [63].

The high potential for reversibility of severe cardiac toxicity of HER2-targeted therapies prompts the implementation of all therapies to sustain hemodynamics, including mechanical circulatory support.

### 2.2. CTLA-4 Blockers and PD1/PDL1 Blockers

Cytotoxic T lymphocyte-associated antigen4 (CTLA-4), also known as CD152, is an inhibitory receptor in the immune checkpoint involved in priming immune responses through the downmodulation of the initial stages of T-cell activation. The CTLA-4 determines a reduction in the immune response in order to reduce damage to healthy tissues. CTLA-4 moves to the T cell surface and concurs with CD28 in the binding to CD80 and CD86, which determines the inhibition of T cell proliferation and activation [64,65,66]. The antitumor action capacity of CTLA-4 blocking monoclonal antibodies was first evaluated preclinically and subsequently validated, even if they have no tumor specificity. Antibody-mediated inhibition of CTLA-4 was the first to report positive results in cancer immunotherapy [67].

The evaluated anti-CTLA-4 blocking monoclonal antibodies were ipilimumab and tremelimumab. These drugs block the CTLA-4 activity and determine the activation of T cells with the onset of the immune response and death of cancer cells. They showed similar properties in patients with advanced solid tumors, with objective response rates of 10% to 15% in patients with metastatic melanoma and renal cell carcinoma [68,69,70].

Ipilimumab was approved for the treatment of patients affected with metastatic melanoma [3] (Table 1). In phase III trials, ipilimumab was administered alone or in combination with a gp-100 peptide vaccine or with chemotherapy dacarbazine, demonstrating superior overall survival compared with the vaccine alone [71] and superior progression-free survival compared with dacarbazine alone [72].

Approximately 20% of patients in both studies achieved long-term survival benefits, suggesting that ipilimumab may induce a state of prolonged disease stabilization [73].

Tremelimumab, in association with imfinzi, was evaluated in patients affected with small-cell lung cancer, metastatic non-small-cell lung cancer, bladder and liver cancer [74,75].

Several adverse events were found after treatment with Ipilimumab, such as fatigue, diarrhea, colitis, myalgias, dermatitis, and hepatitis. Immune-mediated cardiotoxicity, such as myocarditis and pericarditis, has been observed in single patients [76].

PD-1 is a cell surface receptor localized on T, B, and NK cells. PD-1 is also localized on Tregs, NKT cells, activated monocytes, and myeloid DCs. The PD-1, PD-L1, and PD-L2 ligands are expressed on macrophages and DCs [77,78].

In addition, PD-L1 is localized on T and B cells, on cells of vascular endothelium, fibroblastic reticular, epithelium, pancreatic islet, and retinal pigment epithelium cells, and on astrocytes and neurons [77,78,79]. After binding with their ligands, PD-1 receptors inhibit cell proliferation, cytokine secretion, and cytotoxic ability of effector immune cells and reduce the immune response [80]. It has been highlighted that the interaction of CD80 and PD-L1 on APCs that block PD-L1/PD-1 binding determines a reduction in the PD-1 receptor’s function in the early stages of T-cell activation [81,82].

The PD-1/PD-L1 pathway is involved in cancer leak from immunosurveillance; PD-1 was expressed on effector T cells and on exhausted T cells in the tumor microenvironment, while PD-L1 was reported on the cell surface in different cancers, such as breast, bladder, colon, lung, kidney, ovary, melanoma, glioblastoma, and multiple myeloma [78,79].

To date, the therapy that has given the most results in anticancer immune response has been the one that involves the blockade of the PD-1/PD-L1 pathway. The monoclonal anti-PD-L1 antibodies developed to date were Nivolumab, pembrolizumab, and cemiplimab [83] (Table 1).

Nivolumab is a human IgG4 anti-PD-L1 monoclonal antibody that inhibits the interaction between PD-1 and PD-L1. In December 2014, it was approved by the FDA for treating unresectable or metastatic melanoma [4,84], and in March 2015, for treatment of metastatic squamous non-small-cell lung cancer (NSCLC) [64]. Nivolumab has been demonstrated to be effective in several other malignancies, such as Hodgkin’s lymphoma and hepatocellular carcinoma [85]. Nivolumab has an objective response rate (ORR) of 23.7% in patients with NSCLC, whereas the overall survival of patients with Hodgkin’s lymphoma was approximately 80% at three years [86].

Pembrolizumab is a humanized IgG4 kappa anti-PD-1 antibody. In 2014, it was approved by the FDA for the treatment of metastatic melanoma. The ORR of pembrolizumab in patients with advanced melanoma was reported to be 33% [87].

In 2015, advanced NSCLC patients with no previous treatment obtained an objective response rate (ORR) as high as 18%, using pembrolizumab treatment; subsequently, in 2017, pembrolizumab was FDA approved for the treatment of locally advanced or metastatic urothelial carcinoma. Moreover, initial studies reported that pembrolizumab can also be employed in different cancers treatment, a ORR of 53% in non-Hodgkin’s lymphoma and 19% in head and neck squamous cell carcinoma was observed [88].

Cemiplimab is an anti-PD-1 antibody and was recently approved by the FDA. It was the first drug specifically produced for the treatment of advanced cutaneous squamous cell carcinoma (CSCC) [5]. The result of a phase 1 study of cemiplimab treatment in advanced CSCC patients showed a durable response [89].

Its ORR was 47%, with a toxic profile similar to other PD-1 inhibitors.

Three anti-PD-L1 monoclonal antibodies are commercially available, atezolizumab, avelumab, and durvalumab approved by the FDA in September 2014, May 2016, and May 2017, respectively [90].

Atezolizumab is a phage-derived human IgG1 monoclonal antibody with an Fc fragment. Atezolizumab arrests PD-L1 on the surface of the tumor and reports an antitumor capability. In May 2016, it was approved by the FDA as the first PD-L1 inhibitor for urothelial carcinoma [6]. Moreover, Atezolizumab has reported therapeutic effects in other cancers, such as kidney cancer, bladder transitional cell carcinoma, and breast cancer. In patients with metastatic bladder transitional cell carcinoma and breast cancer treated with atezolizumab, a ORR of 26% and 10%, respectively, was observed [91].

Avelumab consists of an anti-PD-L1 IgG1 monoclonal antibody that, by blocking PD-L1, may reactivate T cells and induce antibody-dependent cell-mediated cytotoxicity (ADCC) with its native Fc region. This drug also demonstrated a response of 62.1% in metastatic Merkel cell carcinoma [92]. Moreover, the ORR of advanced NSCLC was 12% [93].

Duravulumab consists of a human monoclonal antibody against PD-L1, that hinders PD-L1 and PD-1 interaction on T cells, resulting in immune responses increase. In neck squamous cell carcinoma (HNSCC) patients treated with duravulumab, an ORR of 9.2% has been reported [79], while in NSCLC patients, 66.3% [7,94,95].

The first immune checkpoint association with anti-PD-1 and CTLA-4 antibodies was in 2009 [87]; a response rate of 60% in patients with metastatic melanoma in phase II and phase III trials was observed, with respect to anti-PD-1 PD-1 blockade alone. Furthermore, it was observed that in the mutated patients treated with PD-1 and CTLA-4, there was a positive response; this suggests that the mutational state could indicate which patients can benefit from this therapy [96]. Therefore, the tumor mutational burden may be useful to identify patients who could take advantage of PD-1 and CTLA-4 blockade combination immunotherapy.

Different studies are focusing on the efficacy evaluation of anti-PD-1/PD-L1 together with anti-CTLA-4 antibodies in the treatment of renal cell carcinoma, mesothelioma, sarcoma, colorectal, lung, esophagogastric, and prostate cancer [83].

#### Cardiotoxicity from Immune Checkpoint Inhibitors

Fatigue, diarrhea, fever, colitis, myalgias, pneumonitis, dermatitis, hepatitis, hypo, and hyperthyroidism are the most common adverse events of PD1/PDL1 blockers treatment. Immune-mediated cardiotoxicity, such as myocarditis and pericarditis, has also been observed [76] (Table 3). There is no evidence that cardiac toxicity correlates with the ICIs dose [97]. Cardiac toxicity is characterized by the presence of ICIs antibodies, which are, in most cases, the IgG isotype. However, some of these reactions involve IgE antibodies; the presence of specific IgE has been shown in patients treated with rituximab [98].

Johnson et al. analyzed a population of patients with metastatic melanoma treated with ICIs who developed myocarditis. They evaluated the T cell receptor on cardiac and skeletal muscle and tumor samples biopsies, reporting an increase in T cells in all tissues. However, no IgG deposition in cardiac or other tissue was found; therefore, the mechanism underlying the reactivity of T cells in the myocardium is not known [99].

As regards the mechanism of ICIs-induced cardiac toxicity, few data are available.

The role of CTLA-4 and PD-1 in determining the peripheral tolerance of the immune system towards self-antigens can be involved. The consequence might be the activation of focal or systemic autoimmune phenomena underlying the clinical picture of immune-related adverse events. Moreover, in animal models, both CTLA-4 and PD-1 demonstrated a protective effect against immune-mediated cardiac damage [106,107].

In patients with suspected cardiac toxicity from ICIs, the diagnosis relies on symptoms, ECGs, echocardiographs, and biomarkers, without peculiar signs. The American Society of Clinical Oncology (ASCO) [108] guidelines graded cardiac toxicity into four degrees: G1, including patients with abnormal ECG or cardiac biomarkers levels; G2, when the above-mentioned findings associate with symptoms; G3, with an onset of symptoms during mild activity; G4, characterized by moderate to severe decompensation, intravenous medication or intervention required, life-threatening conditions. Myocarditis represents an uncommon but clinically relevant manifestation of cardiac toxicity from ICIs, due to its high fatality rate.

In ICI-related myocarditis, LVEF can be normal in a significant proportion of patients but usually depressed in the fulminant syndromes [109]. In the excellent review by Palaskas et al. [110], the role of more advanced diagnostic techniques is summarized. Myocarditis diagnosis is difficult due to the low number of studies.

A specific diagnostic pattern of cardiac magnetic resonance is still not available, and the anamnestic and clinical criteria are essential. Patterns of endomyocardial biopsy are based on specimens from only a few cases [99].

A study [109] evaluated the differences between 35 patients with ICI-associated myocarditis versus a casual cohort consisting of 105 ICI-treated patients without myocarditis. The prevalence of major adverse cardiac events (MACE), such as the composite of cardiovascular death, cardiogenic shock, cardiac arrest, and hemodynamically, significant complete heart block, was evaluated in this multicenter cohort. The prevalence of myocarditis was 1.14%, with a median time onset of 34 days from the first infusion. Combination ICI therapy and diabetes were more common in cases. Forty-six percent of all myocarditis cases experienced a MACE. Myocarditis often showed a fulminant and malignant course. Causes of death included two sudden deaths, two documented ventricular arrhythmias, and two cardiogenic shocks.

High-dose, intravenous steroids were the most commonly administered therapy.

Escudier et al. [111] reported 30 cases of ICI-related cardiac toxicity, with an onset 2–454 days after the first dose (median 65 days). Left ventricular systolic dysfunction was reported in 79% of patients, Takotsubo-like syndrome in 14%, atrial fibrillation in 30%, ventricular arrhythmias in 27%, and conduction disorders in 17%. The cardiovascular mortality rate was 27% due to refractory ventricular arrhythmias, heart failure, pulmonary embolism, and sudden death.

It was observed that with corticosteroid therapy, left ventricular systolic dysfunction can completely recover without evidence of a reduction in the efficacy of the immunotherapy. The fulminant course of the ICI-related myocarditis has been confirmed by Moslehi et al., who reported a 46% mortality rate in 101 cases. In this cohort, the median onset from the first dose was 27 days [112].

As regards treatment, no prospective studies have been conducted. The treatment with ICI must be promptly suspended. On the basis of the available case series, Ganatra et al. [113] suggested a high dose of corticosteroids (i.e., methylprednisolone 1000 mg per day for 3 days followed by prednisone 1 mg/kg) as the first line of therapy in the acute phase. If the patient is unstable, anti-thymocyte globulin, intravenous immunoglobulin, and plasma exchange should be considered. Importantly, there is no evidence that corticosteroids decreased the efficacy of ICIs [114].

Potential alternatives to steroids in the ASCO guidelines include methotrexate, mycophenolate mofetil, azathioprine, rituximab. Infliximab is contraindicated due to its potential to induce heart failure [108]. An excellent meta-analysis on toxic effects of ICIs is found in [115].

### 2.3. Adoptive T-Cell Transfer Therapy

T-cell transfer therapy, also called adoptive cell therapy, increases the capacity of immune cells to attack tumor cells. It includes tumor-infiltrating lymphocytes (or TIL) therapy and chimeric antigen receptor-modified T (CAR-T) cell therapy. Both consist of the recovery of immune cells, in vitro growth, and re-administration to patients. TIL therapy employs tumor-infiltrating T lymphocytes present in the tumor. CAR-T immunotherapy has reported clinical efficacy in blood tumors. The main steps of this therapy are: the harvesting of T cells, their genetic modification to express an antigen receptor otherwise not present, causing the formation of a chimeric molecule, a T cell which reports the specificity like an antibody [8].

CARs increase the T cells’ ability to attack cancer cells by binding to specific proteins present on their surface. The CAR T-cell therapies approved by the FDA for hematological tumors are Tisagenlecleucel, Axicabtagene ciloleucel, and Brexucabtagene autoleucel. Tisagenlecleucel and Axicabtagene ciloleucel are currently FDA-approved for the treatment of B cell acute lymphoblastic leukemia (B-ALL) and diffuse large B-cell lymphoma (DLBCL), respectively [8].

Recently, the FDA approved immunotherapy for some patients with mantle cell lymphoma. The treatment, called brexucabtagene autoleucel (Tecartus), was approved for patients with mantle cell lymphoma unresponsive to other treatments. In a clinical trial, ZUMA-2 was evaluated for treatment with brexucabtagene in 60 patients affected by mantle cell lymphoma who already had more than five previous treatments. Eighty-seven percent of the patients showed an answer to a single infusion, while 62% reported a complete response [116]. CAR T-cell therapy has also been experimentally evaluated in solid tumors treatment, such as breast and brain cancers.

There are some problems in the use of this therapy in solid tumors. In most cases, there are alterations of the cancerous microenvironment of solid tumors. In addition to solid tumors, CAR-T therapy could be used for viral infections, such as HBV infection [117].

#### Cardiac Toxicity and CAR-T Therapy

CAR-T therapy has a meaningful cardiac and systemic toxicity, mainly the cytokine release syndrome, which can lead to a high fever and flu-like symptoms, but also neurologic effects.

In CAR-T-treated patients, cardiac toxicity has been demonstrated by the pivotal randomized controlled trials [118,119,120] (Table 4). There are little data regarding adult patients.

The diagnosis of cardiac toxicity of CAR-T has no specific or additional aspects compared to the detection of general cardiovascular adverse effects [129]. In the context of CRS, hypotension is multifactorial; cardiac contribution to hemodynamic instability has to be detected with echocardiography. Alvi et al. showed that after CAR-T, 54% of 137 patients showed an elevated troponin, but cardiovascular events occurred in 12% [125]. An increase in natriuretic peptides is also usual following CAR-T [122]. Cardiac toxicity can arise in the context of the well-known complications of CAR-T, the cytokine release syndrome (CRS), or, more rarely, can be isolated [130].

CRS can be defined as a cytokine storm. Cytokine storm is at the extreme of the severity spectrum of hyperinflammatory states, and it is characterized by the presence of (i) constitutional symptoms, mainly fever, myalgia, fatigue, and mild hypotension; (ii) elevated levels in the blood of cytokines with a recognized inflammatory effect; (iii) multiple organ failure, if left untreated [131].

The pathogenesis of CRS is related to the occurrence of an uncontrolled release of pro-inflammatory cytokine as a result of the expansion of CAR-T cells. IL-1, IL-6, and TNF-alpha are the major players in the pathogenesis of CRS, and their levels are linked to the severity of the clinical picture. Moreover, the key role of IL-6 is demonstrated by the efficacy of the IL-6 antagonist as a therapeutic measure. The stimulation of receptor-bounded and soluble IL-6 receptors by IL-6 acts through the JAK/STAT transcription pathways.

CRS frequently occurs [129]. In the ZUMA-1 [120], a multicenter clinical trial, 111 patients affected by diffuse large B-cell lymphoma, primary mediastinal B-cell lymphoma, or transformed follicular lymphoma, were enrolled and received CAR-T. CRS was observed in nearly all patients (93%), but most cases were low grade; 13% showed a CRS > grade 3.

The long-term results of ZUMA-1 [121] showed that, after a median follow-up period of 27.1 months, grade 3 or worse CRS occurred in 12 (11%) patients.

In the JULIET trial [119], which enrolled 93 adult patients with relapsed/refractory B-cell lymphoma, CRS was observed 58% of the patient cohort, with 22% of patients showing a severe form.

From a clinical standpoint, fever is always present in CRS, and the time-to-fever interval (the interval between the administration of the therapy and the onset of fever) and the peak temperature have been included in predictive scales.

Several scales for CRS grading have been developed and share the same categorization from grade 1 to grade 4 [132]. Grade 3 and grade 4, as they are characterized by the need for vasopressors and organ damage, warrant ICU admission. Grade 2, if low dose vasopressors or fluid administration are sufficient to ensure hemodynamic stability, can be managed in a medium-intensity ward. The differential diagnosis with sepsis is hard to perform. Sepsis is frequently encountered in patients receiving CAR-T. The correct identification of these two conditions, when they are not concomitant, can determine the patients’ prognosis. Indeed, the immunosuppressive therapy directed to limit CRS could be detrimental in septic patients.

Different plasma profiles of cytokines could help. Indeed Interferon (IFN)-gamma levels are usually more elevated in CRS, whereas interleukin-1β, procalcitonin, and markers of endothelial damage are increased in sepsis. However, the frequent concomitant presence of the two conditions warrants continuous clinical monitoring [131].

The data about cardiac toxicity in adult patients receiving CAR-T are limited.

Alvi et al. [125] conducted a retrospective study to evaluate cardiac toxicity in a population of 137 patients who received CAR-T. The median age was 62 years, and most were male (76%). CRS, occurring a median of 5 days (IQR: 2 to 7 days) after CAR-T, occurred in 59%, and 39% were grade ≥ 2. Twenty-nine patients had their LVEF measured, and in 8 (28%) a decreased LVEF was observed. In the whole population, 17 patients (12%) underwent a cardiac event: six CV deaths, six decompensated heart failure, and five arrhythmias; of note, all events occurred in patients with grade ≥ 2 CRS. The time between CRS onset and tocilizumab administration was associated, for each 12-h delay, with an increased risk of 1.7-fold.

The majority of adverse cardiac events seem to happen early after treatment.

Indeed, Cordeiro et al. [133] reported late adverse events (starting or persisting beyond 90 days after CAR-T infusion) of 86 patients, and no adverse cardiovascular were observed.

In Table 5, the treatment of severe cardiac toxicity from CAR-T is summarized.

## 3. Conclusions

Immunotherapy represented a major advancement in the treatment of solid tumors and lymphoproliferative diseases. These drugs, though heterogeneous from the chemical point of view and their mechanisms of action, share a cardiac toxicity potential. Lessons learned from anthracycline cardiac toxicity made the clinicians aware of the importance of the patients’ clinical surveillance, enrolling them in cardio-oncology programs. Cancer patients sometimes pay the toll of the high toxicity of new therapies in the search for improved survival. The application of intensive care treatment to cardiac toxicity could contribute to the final prognosis, as many patients could recover their cardiac function, then keeping the candidacy to receive further cancer therapies [134]. The increased administration of immunotherapy, driven by positive trials, will unveil the whole picture of cardiac toxicity.

## Figures and Tables

**Figure 1 cancers-13-04797-f001:**
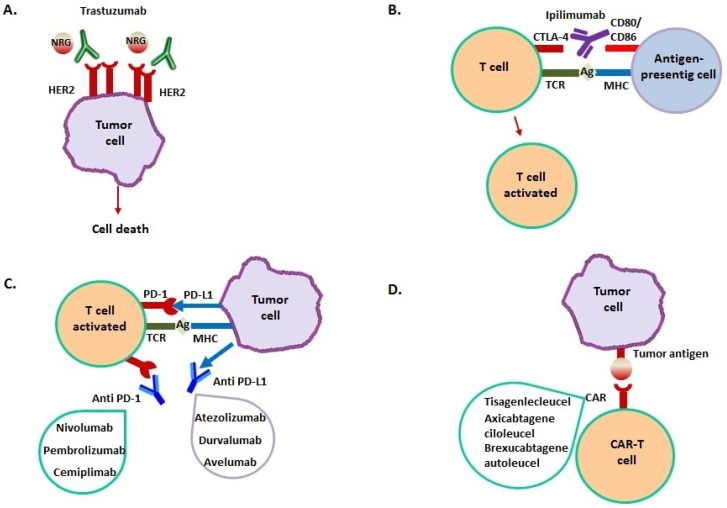
Mechanism of action of immunotherapeutic drugs. (**A**) Trastuzumab binds to the extracellular domain of HER2, overexpressed in breast cancer, inhibiting homodimerization. This prevents HER2-mediated signaling that determines cellular proliferation, then inducing cancer cell death. (**B**) Ipilimumab binds to CTLA-4 on T cell surface, blocking CTLA-4–CD80 or CTLA-4–CD86 interaction. This leads to T cell activation with migration towards their cognate antigen presented by cancer cells. (**C**) PD-L1 is expressed by many cancer cells; it binds to PD-1 on T cell surface, resulting in a suppression of T cell-mediated immune response against cancer cells. Anti-PD-1/PD-L1 monoclonal antibodies block their interaction, enhancing T-cell activity against tumor cells. (**D**) Cancer immunotherapy by CAR-T involves T cells genetic modification to express an antigen receptor that is usually not report. This results in the formation of a T cell chimeric molecule that binds to specific proteins on the cancer cell surface (**D**).

**Figure 2 cancers-13-04797-f002:**
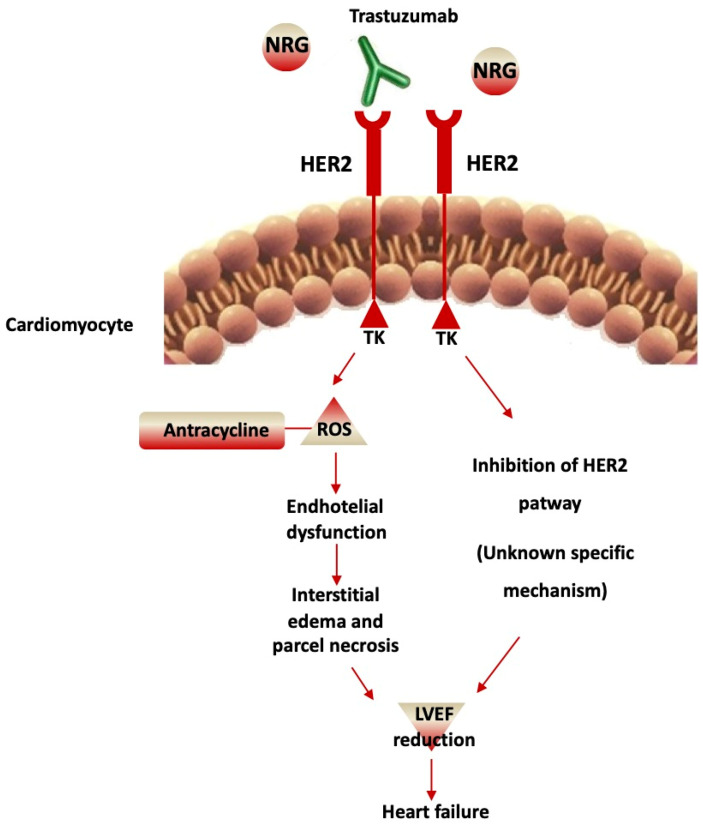
Cardiotoxicity of Trastuzumab. Trastuzumab blocks HER2-mediated signaling through binding to the extracellular domain, which prevents HER2 homodimerization. This leads to the inhibition of the HER2 pathway with an unknown mechanism that induces left ventricular ejection fraction (LVEF) reduction and heart failure. In addition, an increment of reactive oxygen species (ROS) occurs, anthracycline-induced, with endothelial dysfunction. This determines interstitial edema and parcel necrosis, inducing LVEF reduction and heart failure.

**Table 1 cancers-13-04797-t001:** Drugs and main indications.

Drug	Target	Cardiac Toxicity	Tumor Type FDA Approved	References
Trastuzumab	HER-2	Decrease in left ventricular ejection fraction (LVEF)	HER2-positive breast cancer	Keam et al. [2], 2020
Ipilimumab	CTLA-4	Myocarditis	Melanoma	Lipson et al. [3] 2011
Nivolumab	PD-1	Myocarditis	Melanoma Non-small-cell lung cancer Hodgkin lymphoma Head and neck squamous cell carcinoma Urothelial carcinoma Hepatocellular carcinoma	Wei et al. [4], 2018
Pembrolizumab			Melanoma Non-small-cell lung cancer Head and neck squamous cell carcinoma Hodgkin lymphoma Urothelial carcinoma Gastric and gastroesophageal carcinoma	Wei et al. [4], 2018
Cemiplimab			Cutaneous squamous cell carcinoma	Markham et al. [5], 2018
Atezolizumab	PD-L1	Myocarditis; acute myocardial infarction	Urothelial carcinoma Non-small-cell lung cancer	Zhang et al. [6], 2017
Durvalumab		Pericarditis; acute myocardial infarction; atrial fibrillation; cardiogenic shock	Urothelial carcinoma Non-small-cell lung cancer	Lee et al. [7], 2017
Avelumab		Myocarditis; acute myocardial infarction	Merkel cell carcinoma Urothelial carcinoma	Lee et al. [7], 2017
Tisagenlecleucel	CAR T-cell	Decrease in left ventricular ejection fraction (LVEF). Cardiac toxicity as complications of the cytokine release syndrome (CRS)	B cell acute lymphoblastic leukemia	Ahmad et al. [8], 2020
Axicabtagene ciloleucel			Diffuse large B-cell lymphoma	Ahmad et al. [8], 2020
Brexucabtagene autoleucel			Mantle cell lymphoma	Ahmad et al. [8], 2020

HER-2, human epidermal growth factor receptor 2; CTLA-4, cytotoxic T lymphocyte antigen 4; PD-1, programmed cell death 1; PD-L1, Programmed Death-Ligand 1; CAR-T, chimeric antigen receptor-modified T.

**Table 2 cancers-13-04797-t002:** Cardiac toxicity requiring therapeutic interventions secondary HER-2 inhibitors.

Reference	Year	Therapy	No. of Patients	Characteristics and Mean Outcomes
Slamon et al. [30]	2001	Standard chemotherapy vs. standard + Trastuzumab in women with metastatic breast cancer, overexpressed HER2. Follow-up 30 months.	469	63 patients with symptomatic or asymptomatic cardiac dysfunction - 27% of patients receiving trastuzumab + anthracycline + cyclophosphamide - 8% receiving anthracycline + cyclophosphamide - 3% receiving trastuzumab + paclitaxel - 1% receiving paclitaxel alone
Seidman et al. [31]	2002	Trastuzumab	202	Cardiac dysfunction noted in - 27% of patients receiving TRAS + anthracycline + cyclophosphamide - 13% of patients receiving TRAS + paclitaxel - 3–7% receiving TRAS alone Majority of patients with TRAS-related cardiotoxicity (75%) were symptomatic
Geyer et al. [32]	2006	Lapatinib	324	Asymptomatic cardiac events in four patients receiving LAP + CAP vs. one patient receiving CAP alone No symptomatic events and no difference in mean LVEF values between groups
Blackwell et al. [33]	2012	Lapatinib + trastuzumab vs. Lapatinib alone	291	11 patients in the combination arm vs. 3 patients in the monotherapy arm experienced cardiac events 10 events in the combination arm were serious events, including one death
Baselga et al. [34]	2012	Lapatinib + trastuzumab	455	154 women to the lapatinib group 149 to the trastuzumab group 152 lapatinib + trastuzumab - A single patient in each treatment arm experienced decreased LVEF - One patient receiving LAP + TRAS experienced class III CHF (recovered after treatment interruption)
Piccart-Gebhart et al. [35]	2019	Lapatinib + trastuzumab	8381	- 53 patients had symptomatic CHF, including severe CHF (NYHA class II, III, IV) - 18 patients had severe CHF (NYHA class III, IV) - 403 patients had LVEF ≥ 10 decrease and ≥ LLN (based on worst case on therapy) - 97 patients had LVEF ≥ 10 decrease and < LLN (based on worst case on therapy) - Low incidence of primary cardiac events (0.25–0.97% of patients)
Swain et al. [36]	2015	Pertuzumab + trastuzumab	808	- 27 patients (6.6%) of 394 in the PERT group had reduced LVEF - 34 patients (8.6%) of 378 n the placebo group had reduced LVEF Declines were reversed in 21 of 24 patients (87.5%) in the pertuzumab group and 22 of 28 patients (78.6%) in the control group.
von Minckwitz et al. [37]	2017	Pertuzumab + trastuzumab	4805	- 17 patients (0.7%) in the PERT group experienced a primary cardiac event - 8 patients (0.3%) in the placebo group experienced a primary cardiac event - 15 patients in the pertuzumab group and 6 patients in the placebo group had NYHA class III or IV heart failure, and a substantial decrease in left ventricular ejection fraction, and 2 patients in each group died from cardiac causes. - Secondary cardiac events occurred in 64 patients (2.7%) in the pertuzumab group and 67 patients (2.8%) in the placebo group
Verma et al. [38]	2012	Trastuzumab emtansine (T-DM1)	481 T-DM1, 445 lapatinib-capecitabine (LC)	LVEF decline < 50% or below 15% baseline: - TDM-1 8 patients (1.7%) - LC 7 patients (1.6%) LVEF decline < 40% - 3 patients in each group Grade 3 left ventricular systolic - 1 patient in the T-DM1 group - No patients in the lapatinib–capecitabine group.
Krop et al. [39]	2014	Trastuzumab emtansine (T-DM1)	602; 404 to TDM-1, 198 physicians’ choice	LVEF decrease of ≥ 15% from baseline in 1% of patients treated with T-DM1 vs. 1% treated with physician’s choice of therapy
Krop et al. [40]	2015	Trastuzumab emtansine (T-DM1)	153	Asymptomatic LVEF declines (≥ 10 percentage points from baseline to LVEF < 50%): 4 patients (2.7%)
Perez et al. [41]	2017	Trastuzumab emtansine (T-DM1)	1095 a	LVEF decrease of ≥ 15 points from baseline in 0.8% of patients treated with T-DM1 vs. 4.5% treated with TRAS + taxane vs. 2.5% T-DM1 + PERT
Awada et al. [42]	2016	Neratinib	479 patients randomly assigned to neratinib-paclitaxel (n = 242) or trastuzumab-paclitaxel (n = 237)	Grade 3 or higher cardiac events (i.e., cardiac failure, decreased ejection fraction, left ventricular dysfunction and peripheral edema) were reported in three patients (1.3%) in the neratinib-paclitaxel group and seven patients (3.0%) in the trastuzumab-paclitaxel group.
Martin M [43]	2017	Neratinib	2840 (Neratinib 1420, placebo 1420)	Specifics of cardiac safety not reported

CHF, Chronic Heart Failure; NYHA, New York Heart Association; LVEF, Left Ventricular Ejection Fraction; MACE, Major Adverse Cardiac Event.

**Table 3 cancers-13-04797-t003:** Cardiac toxicity requiring therapeutic interventions secondary to immune checkpoint inhibitors (ICIs).

Reference	Year	Therapy	Class	No. of Patients	Patients
Brahmer et al. [64]	2012	Nivolumab	Anti-PDL1	207	1 patient experienced myocarditis.
Horn et al. [100]	2018	Atezolizumab + Carboplatin + Etoposide vs. placebo + Carboplatin + Etoposide	Anti-PD-L1	198 in the ICI arm and 196 in the control group.	1 patient in the ICI arm experienced AV block.
Antonia et al. [95]	2017	Durvalumab	Anti-PD-L1	476 received durvalumab and 234 received placebo	In the ICI group: - 2 patients had acute myocardial infarction - 4 patients had atrial fibrillation - 5 patients had heart failure - 2 pericardial effusion - 1 patient had cardiogenic shock - 1 patient had VT - 1 patient experienced hypertension
Barlesi et al. [101]	2018	Avelumab	Anti-PD-L1	Random assignation to receive avelumab (n = 393) or docetaxel (n = 364).	- 1 acute myocardial infarction in the control group - 1 hearth failure in the treatment group - 1 myocarditis in the treatment group - 4 patients had hypertension in the treatment group and 1 in the control group
Socinski et al. [102]	2018	Atezolizumab plus carboplatin plus paclitaxel (ACP), bevacizumab + carboplatin + paclitaxel (BCP), or atezolizumab + BCP (ABCP)	Anti-PD-L1	393 patients were assigned to the ABCP group, and 394 to the BCP group	- 1 acute myocardial infarction in the ICI group - 1 hearth failure in both groups - 75 patients had hypertension in the treatment group and 67 in the control group
Maio et al. [103]	2017	Tremelimumab	Anti-CTLA-4	382 in the ICI arm and 189 in the control arm	- 3 patients had acute myocardial infarction in the ICI group - 11 patients had atrial fibrillation in the ICI group while 7 had it in the control arm - 2 patients in the ICI group had atrial flutter - 4 patients had heart failure in the ICI group against 2 in the control arm. - 12 pericardial effusion in the ICI group while 6 in the control arm - 2 patients had cardiac arrest in the ICI group.
Robert et al. [104]	2015	Pembrolizumab vs. Ipilimumab	Anti-PD1 vs. Anti-CTLA-4	555 in the ICI arm and 256 in the control arm (ipilimumab)	4 patients had hypertension in the ICI group
Patnaik et al. [105]	2015	Pembrolizumab + ipilimumab (anti-CTLA-4)	Anti-PD1 vs. Anti-CTLA-4	51	1 patient developed myocarditis

CHF, Chronic Heart Failure; NYHA, New York Heart Association; LVEF, Left Ventricular Ejection Fraction; AV, Atrio-Ventricular; MACE, Major Adverse Cardiac Event; VT, Ventricular Tachycardia; ICI, Immune Checkpoint Inhibitors; CTLA-4, Cytotoxic T lymphocyte-associated antigen 4; PD, Programmed Death; PD-L1, Programmed Death-Ligand 1.

**Table 4 cancers-13-04797-t004:** Cardiac toxicity requiring therapeutic interventions secondary to CAR-T.

Reference	Year	Therapy	Patients
Maude et al. [118]	2018	Tisagenlecleucel	Cytokine release syndrome occurred in 58 of 75 patients (77%); the median time to onset was 3 days (range, 1 to 22), and the median duration was 8 days (range, 1 to 36). - 35 of 75 patients (47%) were admitted to the intensive care unit (ICU) for management of the cytokine release syndrome, with a median stay of 7 days (range, 1 to 34). - Nineteen patients (25%) were treated with high-dose vasopressors, 33 (44%) received oxygen supplementation - 10 (13%) received mechanical ventilation - 7 (9%) underwent dialysis - 28 (37%) received tocilizumab for management of the cytokine release syndrome.
Locke et al. [121]	2018	Axicabtagene ciloleucel	108 received Axicabtagene ciloleucel Patients enrolled were 18 years or older. 63 patients experienced hypotension. - 19 patients grade 1 CRS, - 29 patients grade 2 CRS, - 14 patients grade 3 CRS, - 1 patientgrade 4 CRS.
Schuster et al. [119]	2019	Tisagenlecleucel	93 patients received an infusion, CRS occurred in 58% of the patients: 15 patients with grade 3 and 9 grade 4.
Burstein et al. [122]	2018	Chimeric antigen receptor (CAR)-modified T cells targeting CD19 for pediatric acute lymphoblastic leukemia (ALL)	Total patients: 98 24 patients had hypotension-requiring inotropic support with a mean onset of 4.6 days after CAR-T cell infusion - 10 patients (41%) had new echocardiographic evidence of systolic dysfunction - 6 (6%) patients had persistent cardiac dysfunction on discharge echocardiogram 21 of these 24 patients (21%) had hypotension, requiring tocilizumab with or without steroids. No CAR-T cell infusion-related or cardiac-related deaths
Fitzgerald et al. [123]	2017	Pediatric subjects with relapsed/refractory acute lymphoblastic leukemia treated with chimeric antigen receptor-modified T-cell therapy	Total number of patients 39 - 13 had profound fluid-refractory vasoplegic shock treated with α-agonist infusions - 1 had cardiomyopathy with decreased left ventricular systolic function treated with milrinone. Cardiovascular dysfunction developed a median of 5 days after infusion. Shock was catecholamine resistant in 10 of 14 subjects. 13 of 14 subjects with cardiovascular dysfunction were treated with tocilizumab 8 subjects were also treated with short courses of corticosteroids (median, 6.5 d) for refractory hypotension All patients requiring tocilizumab and/or steroids for grade 4 CRS subsequently achieved disease remission and survived CRS.
Porter et al. [124]	2015	CAR-modified T cells to treat 14 patients with relapsed and refractory CLL	29 patients 9 patients with CRS 1 to 14 days (median, 7 days) after CTL019 infusion - anti-cytokine directed therapy in 5 patients a median of 9.5 days after infusion - 4 patients required an intensive care unit (ICU) level of care for complications related to CRS, such as hypotension and hypoxia, with a median length of ICU stay of 6 days
Alvi et al. [125]	2019	CAR-T	137 patients enrolled 55 patients experienced CRS syndrome of at least grade 2. Total of 17 CV events (12%) with a median time to event of 21 days. - 6 CV deaths, - 6 patients with decompensated HF - 5 patients with new-onset arrhythmias.
Burstein et al. [122]	2018	CAR-T	98 subjects hypotension requiring inotropic support occurred in 24 patients with mean onset 4.6 days (range, 1 to 9) after CAR T-cell infusion, including 6 patients receiving milrinone. Worsened systolic function occurred in 10 patients. No cardiac-related deaths.
Hay et al. [126]	2017	CAR-T	133 patients with relapsed/refractory B-cell malignancies 1 patient with grade ≤ 3 CRS developed cardiac toxicity.
Ganatra et al. [127]	2020	CAR-T	187 patients included 12 (10.3%) patients developed new (n = 11) or worsening cardiomyopathy (n = 1), with a decline in median LVEF from 58% to 37% after a median duration of 12.5 days from CAR T-cell infusion. Most patients with cardiomyopathy experienced grade ≥ 2 CRS (11/12) and, as a consequence, were more often treated with tocilizumab, vasopressor support, and mechanical ventilation than those without cardiomyopathy 10% of patients develop cardiomyopathy in the context of high-grade CRS following CAR T-cell therapy
Lefebvre et al. [128]	2020	CAR-T	145 adult patients undergoing CAR-T cell therapy Thirty-one patients had MACE (41 events) at a median time of 11 days - 22 heart failure events (one of which was stress-induced cardiomyopathy) in 21 patients (15%), - 12 episodes of atrial fibrillation in 11 patients (7.5%), - 2 events of other arrhythmias (supraventricular tachycardia, non-sustained ventricular tachycardia), - 2 episodes of acute coronary syndrome - 2 cardiac deaths.

CHF, Chronic Heart Failure; NYHA, New York Heart Association; LVEF, Left Ventricular Ejection Fraction; MACE, Major Adverse Cardiac Event; CRS, Cytokine Release Syndrome; HF, Heart Failure; CV, cardiovascular.

**Table 5 cancers-13-04797-t005:** Treatment of severe cardiac toxicity and CRS due to CAR-T.

Treatment	Indications	Mechanism of Action	Dosage
Vasopressors	Severe hypotension	Alpha-adrenergic receptor agonists	As needed by clinical situation
Inotropes	Cardiac dysfunction, cardiogenic shock	Beta-receptors’ antagonists—PDE inhibitors, calcium sensitizing agents	Adrenaline 0.05–02 mcg/Kg/min Enoximone 5–20 gamma/kg/min Milrinone 0.375–075 mcg/kg/min Levosimendan 0.05–0.1 mcg/kg/min
Mechanical support (IABP, VA ECMO, percutaneous VAD)	Cardiogenic shock refractory to pharmacological therapy	Circulatory support	
Tocilizumab	Severe CRS in patients > 2 years	IL-6 receptor blocker	8 mg/kg every 8 h for a maximum of 4 administration
Siltuximab	CRS refractory to tocilizumab and corticosteroids	Monoclonal antibody directed to IL-6, which prevents its binding with the IL-6 receptor	11 mg/kg three times
Corticosteroids	2nd line in non-responders to Tocilizumab	Pleiotropic genomic and non-genomic anti-inflammatory activity	Dexamethasone 10–20 mg every 6 h. or Methylprednisolone 1000 mg/day
Anakinra	Investigational use	IL-1 antagonist.	100 mg for 5 days
Infliximab	Investigational use	TNF-alpha Ab	
Etanercept	Investigational use	TNF-alpha soluble receptor	
Extracorporeal purification therapies	Cytokine release syndrome	Hemadsorption	

IABP, Intra-aortic Balloon Pump; VA ECMO, Veno-arterial Extracorporeal Membrane Oxygenation; VAD, Ventricular Assist Device; CRS, Cytokines release syndrome; CAR-T, chimeric antigen receptor-modified T; mcg/Kg/min, micrograms per kilogram per minute; mg/kg, milligram per kilogram; IL, interleukin, TNF, Tumor Necrosis Factor.

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
