# Peer review of "Cardiac Toxicity Associated with Cancer Immunotherapy and Biological Drugs"

_cancers, 2021, doi:10.3390/cancers13194797_

Round 1

Reviewer 1 Report

The manuscript by Andrea Montisci et.al entitled Cardiac toxicity associated with cancer immunotherapy and biological drugs provide a broad review of cardiotoxicity linked to cancer patients therapy with anti-Her2 mAb, immune checkpoint inhibitors and CAR-T cells. The authors present and discuss a good collection of papers investigating cardiac toxicity associated with cancer immunotherapy. Overall, the paper is well written and logical. Few suggestions that can make the paper more interesting and useful for the journal readers are below:

  1. In Fig 1 it would be better to use one format (shape and color) to draw tumor cells and another format for T lymphocytes.
  2. The authors do not discuss if there is a correlation between IgG isotype used for therapy and cardiotoxicity
  3. Table 3 would be more informative and easy to understand if the authors include a column indicating specificity of antibody (CTLA-4, PD-1 or PD-L1)
  4. Is there correlation between dose of antibodies used for therapy and observed cardiotoxicity?
  5. Do the corticosteroids diminish the therapeutic effects of antibody injections?
  6. The authors discuss the possible mechanisms of cardiotoxicity after Her2 mAb and CAR-T cells therapy. But there is no information mechanisms of cardiotoxicity after therapy with antibodies against CTLA-4, PD-1 or PD-L1
  7. The mechanisms of action for Tocilizumab and Siltuximab are not shown in Table 5

Author Response

  • In Fig 1 it would be better to use one format (shape and colour) to draw tumour cells and another format for T lymphocytes.

Reply: As requested, we have used one format to draw tumour cells and another format for T lymphocytes.

  • The authors do not discuss if there is a correlation between IgG isotype used for therapy and cardiotoxicity

Reply: We have added the information requested in the text, at the beginning of the paragraph about cardiac toxicity of ICIs. There are limited data regarding the correlation between IgG isotype used for therapy and cardiotoxicity. However, we referenced two papers. 

  • Table 3 would be more informative and easy to understand if the authors include a column indicating the specificity of antibodies (CTLA-4, PD-1 or PD-L1).

Reply: Table 3 has been modified accordingly. 

  • Is there correlation between dose of antibodies used for therapy and observed cardiotoxicity?

Reply: This is an excellent point. To the best of our knowledge, there is no evidence of a relationship between the dose of antibodies and cardiac toxicity. A relationship has been observed only for anti-CTLA4, but not for cardiac toxic events. We have accordingly updated at the beginning of the paragraph about the cardiac toxicity of ICIs.

  • Do the corticosteroids diminish the therapeutic effects of antibody injections?

Reply: Thank you for the interesting suggestion. There is no evidence that corticosteroids can reduce the efficacy of immunotherapy. We added this information (lines 485-486 of the revised manuscript). 

  • The authors discuss the possible mechanisms of cardiotoxicity after Her2 mAb and CAR-T cells therapy. But there is no information mechanisms of cardiotoxicity after therapy with antibodies against CTLA-4, PD-1 or PD-L1.

RepThank you for this important remark. We have added some information, taking into account that limited data are available. 

  • The mechanisms of action for Tocilizumab and Siltuximab are not shown in Table 5

Reply: We modified it according to the reviewer’s suggestion.

Reviewer 2 Report

Cardiac toxicity associated with cancer immunotherapy and bi-2 ological drugs by Montisci et al is an interesting article submitted to Cancers. However, it needs additional information to improve the quality of the manuscript.

Specific comments.

  1. In table one the authors could include the drug induced toxic information on a separate column will be nice.

  1. In Figure 1a, the authors show Trastuzumab binds to HER2 positive breast cancer cells, however the authors mentioned in the abstract that Trastuzumab used in HER2 positive breast, colo-rectal, biliary tract and non-small-cell lung cancers. It will be better to label HER2 positive cancer cells instead of breast cancer cells.

  1. Figure 3. The authors show that Trastuzumab promotes cardiac toxicity, however the author could show some mechanism-based cardiac toxicity will add to the manuscript.

Author Response

Specific comments.

  1. In table one the authors could include the drug induced toxic information on a separate column will be nice.

Reply: As requested, in table 1 we have included a column to report the drug-induced toxic.

  1. In Figure 1a, the authors show Trastuzumab binds to HER2 positive breast cancer cells, however the authors mentioned in the abstract that Trastuzumab used in HER2 positive breast, colo-rectal, biliary tract and non-small-cell lung cancers. It will be better to label HER2 positive cancer cells instead of breast cancer cells.

Reply: As requested, in Figure 1a we have indicated the HER2 positive cell as tumor cells, instead of breast cancer cells.

  1. Figure 3. The authors show that Trastuzumab promotes cardiac toxicity, however the author could show some mechanism-based cardiac toxicity will add to the manuscript.

Reply: According to the reviewer’s suggestion, we added more information about the mechanisms underlying the cardiac toxicity of trastuzumab.